# A Systematic Review on Privacy-Aware IoT Personal Data Stores

**DOI:** 10.3390/s24072197

**Published:** 2024-03-29

**Authors:** George P. Pinto, Praveen Kumar Donta, Schahram Dustdar, Cássio Prazeres

**Affiliations:** 1Institute of Computing, Federal University of Bahia, Canela 40170-110, Brazil; prazeres@ufba.br; 2Federal Institute of Bahia, Santo Antônio de Jesus 44571-020, Brazil; 3Distributed Systems Group, TU Wien, 1040 Vienna, Austria; pdonta@dsg.tuwien.ac.at (P.K.D.); dustdar@dsg.tuwien.ac.at (S.D.)

**Keywords:** personal data privacy, internet of things, personal data store, systematic review

## Abstract

Data from the Internet of Things (IoT) enables the design of new business models and services that improve user experience and satisfaction. These data serve as important information sources for many domains, including disaster management, biosurveillance, smart cities, and smart health, among others. However, this scenario involves the collection of personal data, raising new challenges related to data privacy protection. Therefore, we aim to provide state-of-the-art information regarding privacy issues in the context of IoT, with a particular focus on findings that utilize the Personal Data Store (PDS) as a viable solution for these concerns. To achieve this, we conduct a systematic mapping review to identify, evaluate, and interpret the relevant literature on privacy issues and PDS-based solutions in the IoT context. Our analysis is guided by three well-defined research questions, and we systematically selected 49 studies published until 2023 from an initial pool of 176 papers. We analyze and discuss the most common privacy issues highlighted by the authors and position the role of PDS technologies as a solution to privacy issues in the IoT context. As a result, our findings reveal that only a small number of works (approximately 20%) were dedicated to presenting solutions for privacy issues. Most works (almost 82%) were published between 2018 and 2023, demonstrating an increased interest in the theme in recent years. Additionally, only two works used PDS-based solutions to deal with privacy issues in the IoT context.

## 1. Introduction

Recent decades have witnessed a rapid intensification in the digitalization of citizens’ daily lives, both public and private, along with an increase in the generation and collection of large amounts of data, whether personal or not. Big companies such as Google, Facebook, and Twitter have developed by exploiting data and offering free services to users in return. This data-driven approach has started a novel economic paradigm that has proven invaluable in supporting diverse types of businesses [1]. However, despite the benefits of these technologies, it is imperative to realize that malicious people or organizations can exploit personal data to influence behaviors, societies, and even political viewpoints, among other aspects.

Currently, the centralized web structure collects, stores, processes, shares, and negotiates personal data without the true owners’ knowledge or consent. Their magnitude, accuracy, and usage details have been unknown to citizens and governments [2]. Sharing data with websites, social networks, and devices may lead users to lose control over its storage and dissemination, preventing them from managing or modifying their shared information [3]. Furthermore, this centralized structure exposes personal data to privacy concerns. For instance, the exploitation of individuals’ personal data without their consent has become news with scandals such as Cambridge Analytica, which used personal data for political purposes; Polar’s fitness app, which revealed the location of U.S. military and security personnel; and Google Plus, which exposed the names, email addresses, occupations, and ages of 52.5 million users [4]. Such events violate Westin’s privacy definition. He states that privacy is “the right to select what personal information about me is known to what people”, emphasizing the user’s right to control [5]. As such, users have been claiming their privacy protection and looking for ways to control their data and ensure its protection [6].

At the same time, the advent of IoT technologies has enriched our daily lives by enabling interconnected devices to generate and collect an increasing amount of data and enabling the creation of more personalized and valuable services [7]. The IoT encompasses a union of diverse technologies of sensors, networks, communications, computing, and semantics “connecting people and things anytime, anyplace, with anything and anyone, ideally using any path or network and any service” [8]. While the IoT promotes many benefits in our everyday lives, from financial transactions to personal communications, it makes personal information even more vulnerable to access by unauthorized third parties worldwide. Its technologies and characteristics have the potential to amplify privacy issues, posing a trade-off between the convenience of the technology’s diverse services and users’ privacy concerns [9].

Punagin and Arya [10] argue that while safeguarding privacy results in costs to online services in terms of quality and usefulness, user control over data sharing is ultimately essential. Therefore, it is crucial to balance privacy protection and service quality, as restrictions on information access can affect service usefulness. In turn, Alsheikh [11] considers it a misconception to understand privacy preservation as an impediment to innovation, conferring an inaccurate understanding of the data privacy concept. Instead, he advocates user control over data while promoting innovation.

Addressing this privacy challenge requires technological innovations that allow users to regain control over their data [12]. This can be achieved by promoting separation between data storage and services and enabling users to determine where and how their data are stored when accessing the required services. Thus, the concept of Personal Data Stores (PDS) exemplifies this technological effort to leverage users’ control over their data. A PDS serves as a private data repository, with the aim being to deal with user data control and privacy issues by employing a decentralized approach to data processing. The PDS concept represents a paradigm shift in the relationship between citizens and service providers, placing individuals at the forefront. This decentralization underscores the importance of user choice, giving individuals the authority to determine the destiny of their data [13].

In this context, a literature review is an important tool to promote a better understanding of a specific research area, offering insights into its current state and revealing potential areas of exploration. In this study, we conducted a structured and controlled systematic mapping review [14] to identify, evaluate, and interpret the relevant literature associated with privacy issues and PDS-based solutions in the context of the Internet of Things.

Many previous studies have reviewed privacy problems in the IoT context, focusing on different perspectives. A number of these studies have aimed to provide an understanding of the problem in general IoT applications [15,16,17,18,19]. In contrast, others have concentrated on specific IoT environments such as smart farming [20], the Internet of Medical Things [21,22], the Internet of Healthcare Things [23], and the Internet of Industrial Things [24]. Conversely, other authors have dedicated their work to reviewing studies exploring diverse privacy protection solutions. For instance, reviews can be found dedicated to blockchain or machine learning-based solutions, as well as to privacy-enhancing technologies in general. Although previous studies (see Section 2) have reviewed privacy aspects in different IoT environments and presented diverse solutions, to the best of our knowledge they do not address all of the aims proposed in this work. We aim to understand how these privacy issues have been explored and discussed in the IoT literature and how PDS has been adopted to solve these issues.

This article makes significant contributions in the following ways:We describe results and analyses about each of the three questions guiding this study, offering valuable insights that can guide future research;We summarize and review commonly encountered privacy issues (identification, localization and tracking, profiling, privacy-violating interaction and presentation, lifecycle transitions, inventory attacks, linkage, and information leakage) highlighted by the authors, providing a concise overview of the key concerns;We ave grouped the findings related to privacy in the IoT context into four categories: overview or problem identification, solution proposals, awareness or perception, and regulation;Our findings reveal that the most significant number of privacy-related works was published in the last five years, indicating an increased interest in the theme;We position the role of PDS technologies as a solution to privacy issues in the IoT context. The only two solutions presented thus far both rely on a user-centric approach based on Solid to address privacy issues, pointing to the need for more research in this area;Finally, we furnish the reference list for the 49 primary studies, establishing a comprehensive knowledge base for future researchers delving into the field of IoT privacy.

The remainder of this article is organized as follows: Section 2 presents the related works; Section 3 describes the steps performed in conducting our systematic review; Section 4 summarizes and discusses our review results; Section 5 deal with threats to validity; in Section 6, we explore the open challenges of privacy in the IoT context; finally, Section 7 summarizes our contributions and presents potential future work.

## 2. Related Works

In recent years, several literature reviews have been published addressing privacy concerns in the IoT context. Table 1 offers a comparison of the six research papers analyzed in this section. To contrast our proposal with these state-of-the-art works, we evaluated the following attributes: ‘Year’ indicates the publication year of the study; ‘Systematic’ denotes whether the study followed a systematic process; ‘Domain’ specifies the field in which the review was conducted; and ‘Main Goal’ outlines the primary objective of each paper. Below, we offer a concise summary of the chosen studies.

Islam et al. [17] undertook a systematic mapping review aimed at comprehending privacy objectives concerning IoT and Big Data, identifying the various types of privacy attacks that breach these objectives, and evaluating existing solutions designed to mitigate such attacks. The search encompassed studies published between 2010 and 2021. As a result, they categorized privacy goals into three groups: data privacy, which involves keeping private and sensitive data safe from third parties; location privacy, which pertains to the safeguarding of individuals’ locations; and identity privacy, which protects individual identities from third parties. Regarding privacy attacks, their research revealed that inference, membership inference, linkage, and conclusion attacks are the predominant types documented in the literature. Finally, the most available privacy measures were differential privacy, k-anonymity variation, sanitization and randomization, and cryptographic techniques.

Torre et al. [18] performed a systematic mapping study to explore the role of privacy-preservation (PP) techniques for IoT devices. Their findings comprise (i) the existing PP techniques and supporting tools, (ii) the goals and the IoT layers and devices covered by the studies, (iii) the most commonly discussed privacy threats and attacks, and (iv) the often-used evaluation metrics in the area.

Zainuddin et al. [16] conducted a literature review of existing works addressing privacy threats in the context of IoT applications. They identified and discussed various privacy threats, including data leakage, impersonation, data tampering, eavesdropping, and jurisdictional risks. A subsequent discussion delved into these threats across different IoT applications, such as smart homes, smart cities, smart meters, etc.

In their work, Sarwar et al. [25] reviewed the research on privacy preservation in IoT and fog-enabled IoT. They categorized publications into two groups according to the privacy needs of IoT-based applications, namely, content and context. Content privacy was classified into six categories: data aggregation, data querying, behavior and action, media, social interaction, and state of mind and body.

The main objective of Ogonji et al. [15] was to provide an extensive examination of the current IoT literature emphasizing privacy and security risks, attack surfaces, vulnerabilities, and corresponding countermeasures. They conducted a systematic literature review and identified numerous security and privacy issues. IoT security risks encompassed eavesdropping, spoofing, RF jamming, sybil attacks, sinkhole attacks, man-in-the-middle attacks, denial of service (DoS) attacks, malicious code injection, sniffing attacks, and spear-phishing attacks. In parallel, privacy risks involved identification, location and tracking, profiling, interactions, and presentations. Consequently, they introduced a taxonomy of risks addressing IoT privacy and security.

Alabdulatif et al. [24] presented a comprehensive review of the Internet of Nano-Things (IoNT), concentrating on its architectural components, security issues, and privacy challenges. Concerning privacy, their paper explored threats such as data collection, data security, location tracking, surveillance, regulatory gaps, confidentiality, integrity, availability, and authenticity. In addition, it delved into challenges such as resource constraints, internet exposure, key management, secure localization, encryption gaps, and malware risks. Furthermore, their paper classified privacy attacks into different categories such as disruption, disclosure, deception, and usurpation.

In [26], the authors provided a comprehensive review of privacy protection in IoT, focusing on privacy-enhancing technologies (PETs) and GDPR-specific principles. They highlighted three primary privacy challenges in IoT: (1) technical expertise deficiency in privacy notice comprehension; (2) transparency and control issues regarding personal data; and (3) the absence of personalized privacy recommendations.

Zarandi et al. [20] offered a comprehensive examination of Big Data privacy issues in smart farming. Their analysis delved into privacy challenges and requirements across each stage of the data lifecycle while also surveying current solutions and cutting-edge technologies aimed at enhancing data privacy within smart farming.

Demertzi et al. [27] reviewed the literature on the privacy requirements of the Industrial Internet of Things (IIoT). They provided an overview of this area, including its advantages, disadvantages, challenges, and primarily privacy issues. Their focus was on addressing privacy requirements, with particular attention to the processing of personal data by competent authorities. Additionally, their study aimed to provide insights into the challenges and strategies around maintaining industrial privacy within the context of IIoT ecosystems.

Fallatah et al. [28] surveyed the literature on PDS, offering a comprehensive review of related concepts and anticipated benefits. They thoroughly explored the advantages and disadvantages of PDS technology as an alternative solution for a user-centric model that can facilitate individuals’ reclaiming control over their personal data. Finally, they compared and analyzed existing PDS platforms, summarizing the challenges and issues hindering the development and widespread adoption of PDS platforms.

In [29], Zubaydi et al. systematically reviewed the latest advancements around integrating blockchain technologies (BCT) and IoT with a focused approach to addressing security and privacy concerns. They outlined the principles of BCT and IoT, covering architecture, protocols, consensus algorithms, distinguishing features, and integration integration. Finally, they presented solutions to address privacy and security issues, followed by categorizing investigated applications based on various characteristics such as primary information, objectives, development level, target application, blockchain type, platform, etc. In turn, Asqah and Moulahi [30] presented a review of federated learning and blockchain integration within the IoT ecosystem. They reviewed the security and privacy challenges arising from this integration and discussed and categorized solutions for these concerns.

Kamalov et al. [22] conducted a comprehensive analysis of security concerns in the IoT and Internet of Medical Things (IoMT) from various perspectives. They categorized their findings based on the type of application, year of publication, diversity of applications, and other innovative viewpoints. According to their research, blockchains can addressed numerous challenges in security, authentication, and maintenance of IoT systems. In turn, Hireche et al. [21] reviewed the current security and privacy challenges in the IoMT. Their findings revealed a proliferation of methodologies aimed at securing IoMT devices, concentrating on securing the network layer of the device or the body. Proposed solutions for securing these devices encompassed a spectrum ranging from device authentication and sensor anomaly detection to access control measures. Moreover, their research underscored the superiority of blockchain technology, the ECC algorithm, and lightweight authentication mechanisms over traditional algorithms in bolstering security. Finally, they asserted that conventional ML techniques may not be efficient enough when certain metrics are not considered.

In their analysis, Shahid et al. [23] presented a detailed IoHT classification and categorized various healthcare devices according to their operational functions and deployment scenarios. They identified several potential vulnerabilities leading to data breaches, including legal conflicts, utilization of low-quality devices, insufficient awareness, and the absence of dedicated local enforcement authorities. In addition, they compared several data protection regulations and highlighted their limitations. Finally, they provided recommendations related to data privacy and security for IoHT implementations.

Rodriguez et al. [31] examined solutions for privacy in IoT based on machine learning (ML) and deep learning (DL). They identified and categorized privacy threats and challenges hindering the seamless integration of ML into privacy safeguarding measures, and pinpointed solutions tailored to distinct threats and attacks. Their analysis underscores that while privacy-protecting solutions are in the initial stage, achieving superior tradeoffs between privacy guarantees and computational cost remains challenging.

Khan et al. [19] provided an overview of IoT and its technologies and architecture, focusing on features of critical applications such as smart homes, smart agriculture, smart transportation, and smart healthcare. They discussed possible attacks affecting data and infrastructure at different IoT layers. Lastly, they indicated that the security and privacy of data are most critical during the data dissemination phase.

In conclusion, our search revealed numerous works delving into privacy issues within the IoT context; nonetheless, we found a scarcity of prior research addressing our primary objective, that is, to comprehend how these issues have been explored and addressed while establishing the role of PDS in formulating solutions, as described by the research questions in Section 3.1.1.

## 3. Research Method

The systematic mapping process we applied was based on [32,33]; as illustrated in Figure 1, this process is delineated into three main phases: planning, conducting, and reporting.

### 3.1. Planning

The planning phase is a critical and foundational step in the research process, embodying the definition of a review protocol and serving as the cornerstone of the entire review. It sets the stage for the review by establishing clear objectives, defining the scope, and creating a structured framework for conducting the study [14]. Here, we describe the key aspects of the planning phase, outlining the protocol and its steps.

#### 3.1.1. Research Questions

In this step, we define the research questions and objectives of the review which guided the entire process. This study aimed to investigate the current state of solutions for privacy protection in IoT. Specifically, we focused on understanding the use of PDS to deal with privacy threats within the IoT context. Consequently, we prioritized comprehending the roles played by PDS in the IoT ecosystem. To this end, we formulated the research questions (RQs) outlined in Table 2.

The advent of IoT has introduced new privacy issues and amplified existing ones. RQ1 was formulated to uncover these issues, understand the proposed solutions in each case, and identify the concerns that require attention. RQ2 aimed to identify works where PDS was applied within the IoT domain to address various problems. Additionally, we sought to discover and understand existing PDS-based solutions for privacy within that domain. These adopted solutions and their design can serve as references for addressing new privacy challenges. Finally, RQ3 aimed to identify works utilizing PDS to address privacy issues across various contexts. Through an analysis of the results, we sought to comprehend how these works leveraged PDS to resolve privacy concerns and to apply these ideas in other contexts, including IoT.

#### 3.1.2. Search Strings and Sources

The next step is to provide the means to answer the previously mentioned research questions. To achieve this, we extracted the following keywords from them: internet of things, IoT, privacy, fog computing, edge computing, risk, threat, personal data store, personal data vault, social linked data. As outlined in Table 3, we formulated three distinct search strings using the specified keywords. Subsequently, we manually conducted searches in the selected databases presented in Table 4. The selection of these digital libraries was driven by their relevance in the field of computer science [33].

#### 3.1.3. Selection Criteria

As the initial set of studies encompasses irrelevant results, to effectively address research questions it is necessary to discard these irrelevant entries and ensure the inclusion of only suitable studies. To this end, we established the following selection criteria:

**SC1**: The study must be written in English;

**SC2**: The study must be a conference paper, journal article, or book chapter;

**SC3**: The selected study must be available on the web;

**SC4**: The study must be unique, that is, any duplicate entries need to be removed;

**SC5**: The study should present initiatives related to the topic of the research strings.

We selected all available full and short papers published in peer-reviewed journals, conferences, symposia, workshops, or books. We avoided any restrictions in this sense and focused on using the defined criteria in both cases. Additionally, we did not constrain the time frame for publication years in our study. All studies encompassed within our results spanned up to the date of search execution.

### 3.2. Conducting

The conducting phase seeks to identify relevant primary studies related to the research scope. We carried out this study in August 2023, and we obtained the initial set of potentially relevant primary studies by performing search strings in all selected research databases. We applied each search string to the studies’ titles, abstracts, and keywords. As a result (Table 5), we obtained 136 studies from SS1, which was the largest of all the strings. SS2 returned 14 studies, while SS3 returned 26.

Next, we removed all studies that did not match the previously defined selection criteria. After applying criteria SC1 to SC4, we obtained the results depicted in Table 6. We found 85 results in SS1, 12 in SS2, and 22 in SS3. Finally, we applied the SC5 criterion and selected those studies related to our research objectives. For this, we read the papers’ titles and abstracts, and when necessary their introductions and conclusions. In summary, we finished this review phase with 37 studies selected from SS1, seven studies from SS2, and seven studies from SS3.

### 3.3. Reporting

After acquiring the final set of publications, we developed a classification scheme aligned with the research scope and questions delineated in the planning phase (Section 3.1). In our report, we aimed to organize the data in a way that would enable us to interpret them. Thus, we grouped the results by the search string, published year, and main study objectives.

We initiated data organization based on the search results. As depicted in Figure 2, the distribution of studies illustrates the dominance of SS1, with the highest publication rate at 72.5%, surpassing both SS2 and SS3, which each contributed 13.7%. It is notable to highlight the significant volume of publications from SS1, which is nearly three times higher than the combined total of the others. This underscores the heightened emphasis on privacy in IoT.

Second, we considered the publication year. Figure 3 shows a pie chart illustrating the distribution of publications by year. Most retrieved studies were published between 2018 and 2023 (82.4%). Turning to Figure 4, it is evident that approximately 86% of the studies returned from SS2 and SS3 were published between 2020 and 2023.

Lastly, we categorized the studies by considering their primary objectives. We conducted the SS1 search and classified the results into four groups based on the main focus of the research. As depicted in Figure 5, these groups were as follows:

Overview or Problem Identification: This category encompasses studies dedicated to identifying and presenting privacy issues within the context of IoT or fog computing.Solution Proposals: This group includes studies that offer general solutions for addressing different privacy concerns.Awareness or Perception: This category concentrates on studies that address privacy issues as problems related to user awareness or perception.Regulation: Studies in this group raise concerns about privacy and privacy protection from the perspective of laws and regulations.

As depicted in Figure 5, the majority of the retrieved studies (46%) offered an overview or presentation of diverse privacy issues within the IoT context. Approximately 27% proposed solutions to address privacy concerns in IoT, while 18.9% delved into studies focusing on user awareness and perception of personal data privacy. A further 8.1% of the studies examined privacy from a regulatory or legal perspective.

The findings from SS2 (Figure 6) reveal that 57% of studies relating to the combination of IoT and PDS predominantly concentrated on data security and privacy. The remaining research was divided between decentralized data processing (29%) and IoT data control and interaction with PDS (14%).

Finally, in SS3, a noteworthy 85% of the works associated the use of PDS in the privacy context with enhancing users’ control over their data.

## 4. Results

In the previous section, our data analysis provided an objective account of the number of publications considering the search strings, their evolution over time, and the main study objectives. This section delves deeper into the results in order to comprehensively address the research questions. Table 7 depicts all of the selected studies organized by their publication year in ascending order.

As can be noticed, the SS1 search string retrieved most of the papers in our review. These findings highlight a substantial volume of research dedicated to addressing privacy concerns in IoT contexts. Privacy is a recurring problem and is gaining increasing importance, mainly due to the integration of smart devices into people’s daily lives. Furthermore, as observed in the retrieved studies, the past five years have witnessed an increased number of publications, indicating growing interest into researching IoT privacy solutions. In contrast, when examining the outcomes of the other search strings it becomes evident that there has been only limited exploration of the connection between IoT and PDS as a method for addressing privacy risks. Thus, this represents a research area in need of further research and development.

Another point to highlight is the significant portion of works excluded from the SS1 search string that, despite claiming to address the topic, actually proposed security solutions rather than explicitly addressing privacy. As stated in [11], “data privacy may not be met even when the original data is securely stored”. Fulfilling the security principles of confidentiality, integrity, and availability (CIA principles) does not necessarily ensure the protection of privacy. Privacy entails safeguarding users from violations and misuse, which includes how service providers utilize and process user data. Privacy is about user control over their data. For instance, even data stored securely in compliance with CIA principles may still face the identification privacy risk when the data are released. Thus, in this mapping study we considered only studies that explicitly treated privacy, and discarded works relating only to security issues such as cryptography, authentication, authorization, and integrity. In all, 47% of the studies removed according to the SC5 criterion proposed security solutions instead of data privacy proposals.

Following, we used the findings to answer the initial research questions.

### 4.1. Privacy Focus in IoT Studies

In addressing research question RQ1 (see Section 3), our main objective was to identify the most studies on privacy in IoT and to understand their research focuses. Reading and evaluating the selected studies allowed us to answer this question by categorizing these works into four groups: Overview or Problem Identification, Solution Proposals, Awareness or Perception, and Regulation, as outlined in Section 3.3. Each of these groups directs attention towards a distinct focal point in the realm of data privacy research within the IoT domain.

In the Overview or Problem Identification group, we found studies that referenced privacy within IoT without necessarily addressing or defining specific issues (P15 [48], P22 [55], P43 [75]). Instead, these studies emphasized a broad perspective on privacy issues and how IoT has impacted them. In contrast, most of the studies we analyzed aimed to identify privacy risks and challenges in an IoT context. Table 8 illustrates the most frequently identified risks (identification, information leakage, interaction and presentation, inventory attack, lifecycle transitions, linkage, localization and tracking, and profiling) and where in the literature they can be found. These studies identify possible risks and offer concise definitions of each.

In the context of the Solution Proposals group, our findings revealed various approaches aimed at addressing distinct privacy challenges within the IoT context.

P9 [42] introduced a privacy-preserving method for managing access to sensitive IoT data. This technique involves breaking down sensitive data into components stored across multiple data repositories. When users request IoT data, they (re)aggregate it without revealing anything beyond abstract components. The authors also presented an architecture for preserving of privacy that ensures end-to-end privacy.

P11 [44] presented the SPRINT framework to assess and monitor IoT systems’ service security and privacy based on recognized professional standards. In turn, P12 [45] described the impact of 5G evolution on privacy in the IoT and presented a privacy assessment methodology based on the combination of two other frameworks, which aims at the discovery of privacy threats.

The authors of P13 [46], claiming that blockchain-based solutions can preserve IoT privacy, introduced a methodology aimed at mitigating privacy risks from IoT-enabled devices within a smart home environment through blockchain-based smart contracts.

P18 [51] introduced DISPEL, a platform for distributed privacy management in the IoT domain. DISPEL represents a Privacy-by-Design methodology for IoT, striving to offer privacy safeguards from the outset. Through DISPEL, data owners delineate which data an IoT platform is authorized to share with an IoT application and for what purposes.

P27 [60] introduced the creation of a tool-supported theoretical framework that includes a privacy policy language and a model for analyzing IoT systems, aimed at safeguarding user data within IoT environments.

The authors of P23 [56] suggested employing semantic web technologies to handle dynamic privacy choice configuration in IoT environments. The essence of their proposal comprises (i) an ontology for privacy choices within the IoT context and (ii) an interactive Privacy Preference Model along with its corresponding ontology, integrated with a Personal Data Manager aligned with GDPR objectives.

P25 [58] introduced the Heterogeneous IoT Privacy Architecture (HIPA), a framework designed to address privacy issues in diverse environments. This proposed framework has two primary objectives: first, it provides various methods to assess and detect inconsistencies and privacy breaches of a new device prior to its installation; second, it prioritizes customer preferences.

P33 [66] introduced an approach to assessing the tangible impact of IoT privacy breaches on mobile network data. This method integrates examinations with empirical data, establishing privacy markers derived from network activity and linking them to a privacy evaluation framework. Finally, P36 [68] focused on solving the privacy leakage risk, employing a federated learning-based intrusion detection system to differentiate between normal and abnormal behaviors and assessing its efficacy in identifying attacks. The authors also developed a federated deep fuzzy rough convolutional neural network model using neuroevolution (F-DFRCNN-NE) to address IoT privacy and security challenges.

In P46 [78], the authors discussed the common practice of publishing security and privacy policies in text and how difficult it is to understand their content. Regarding this, they propose a knowledge elicitation methodology from textual information to provide an analysis tool for policy analysts, smart city administrators, and IoT vendors to evaluate IoT privacy policy documents according to the National Institute of Standards and Technology (NIST) cybersecurity and privacy framework.

Our findings, as classified within the *Awareness or Perception* group, underscore the importance of understanding individuals’ levels of awareness and their perceptions. This is particularly relevant in the realm of IoT when considering privacy risks.

P2 [35] aimed to empower users in estimating the associated risk of sharing personal data by introducing a privacy management scheme that detects and analyzes sensitivity, resulting in quantification of privacy content through statistical disclosure control and an information-theoretic model.

In P10 [43], the authors conducted interviews with smart home owners to explore the motivations behind IoT device purchases, perceptions regarding privacy risks in smart homes, and measures taken to safeguard privacy. Their results suggest that users prioritize convenience and connectivity, which shape their attitudes and actions concerning privacy. Users grant access to their smart homes depending on perceived external benefits. Additionally, users tend to believe their privacy is safeguarded due to their trust in IoT device manufacturers. However, they must be aware of the potential for machine learning inference to expose sensitive data. These findings contribute to developing a prototype privacy and security label that can help consumers to make better-informed decisions when purchasing IoT-related products.

P21 [54] conducted a questionnaire-oriented survey with two main objectives: (i) assessing users’ inclination towards controlling data disclosure and (ii) preference for transparent privacy preservation based on users’ needs in their homes. Their results reveal that users are cognizant of data collection and its privacy implications, and perceive themselves as capable of managing data collection. Subsequently, they established a set of conditions, termed UCCPs, designed for smart homes and user privacy preservation. P39 [71] also conducted a questionnaire-oriented survey to explore individuals’ understanding of privacy in order to enhance awareness of privacy protection for data users and organizations. Their findings reveal moderate awareness of information privacy protection among IoT users, suggesting a need for more education and awareness. P44 [76] carried out a questionnaire-based study with respondents from Saudi Arabia in order to investigate their understanding of the privacy risks of using IoT devices. Their main findings reveal that users believe that IoT devices affect their privacy.

Lastly, the authors of P40 [72] aimed to explore users’ perceptions of the sensitivity levels of various information collected by IoT devices and their willingness to share this information with third parties. To this end, they conducted a study among students and workers recruited from Amazon Mechanical Turk. Participants were presented with scenarios involving IoT devices that collected information and asked to assess their level of awareness and their willingness to divulge the information. Consequently, they noted that users exhibited varying sensitivity levels which were influenced by their gender, and that the inclination of users to divulge their own data was contingent upon the sensitivity level of the information and the categories of the third parties involved.

From the perspective of the *Regulation* group, we obtained results highlighting the concern ariund and importance of privacy protection as well as the primordial role of regulation.

P4 [37] expressed concern about individual privacy protection and underscored the imperative need for appropriate regulations. At the time, existing laws focused only on basic data protection and failed to adequately address the intricate challenges inherent in the IoT landscape.

In P45 [77], the author discussed privacy and ethical concerns around smart meters, exposing privacy risks associated with smart meters from the GDPR perspective and highlighting the complexity of these concerns and the importance of compliance with GDPR requirements when designing an IoT system.

Finally, P41 [73] presented findings from a set of interviews with stakeholders from different areas, focusing on understanding their vision related to privacy in Australia. The findings outlined that the range of concerns goes beyond established issues about security and privacy, expanding to others such as impacts on vulnerable communities, the environment, etc. Stakeholders expressed the view that more robust regulation is required in Australia.

### 4.2. Issues Addressed by PDS in the IoT Context

In addressing research question RQ2 (see Section 3), we acquired seven studies: P29, P30, P31, P42, P47, P35, and P49. In response to this question, we can highlight the small number of works that proposed PDS-based solutions to solve problems in the IoT domain. Two of the solutions we found applied PDS to address privacy challenges, while the others focus on different challenges, as observed in Table 9. We describe each study in detail below, highlighting and comparing them with a focus on (i) the IoT challenge the proposal aims to solve; (ii) the implementation of PDS used in the proposal; and (iii) the IoT scenario in which the solution was applied.

P30 [63] proposed PDSProxy to achieve confidential processing for personalized AI services on untrusted third-party IoT devices by enabling personal data secure transmission across hierarchically operating nodes. Additionally, this solution facilitates personal data caching and streaming over trusted nodes in hierarchical systems. In a subsequent study, P29 [62], the same authors addressed the issue of high initialization overhead by introducing PDSProxy++, a PDS extension designed for proactive AI services deployment on nearby IoT devices.

In P31 [64] the authors introduced SOLIOT, an approach merging concepts from Linked Data and Solid, to demonstrate how web technologies can address IoT challenges such as constrained devices, battery-powered sensors, and network disruptions. Their primary aim was to resolve issues in industrial applications by bridging the gap between the web and local manufacturing environments. They achieve this by integrating lightweight industrial protocols and enhancing data control through Solid. SOLIOT presents a resource-driven view that simplifies consumption and facilitates interaction with digital twins.

P42 [74] presents SHARIF, a novel concept that merges the Solid ecosystem with blockchain technology to establish a decentralized solution for healthcare data communications. This approach is proposed as an alternative to traditional database management systems (DBMS), which are plagued by significant issues. The primary objective is to tackle security concerns inherent in conventional healthcare data systems by leveraging smart contracts, thereby creating a secure patient-centric framework for data exchange. The authors assert that this proposal can help to alleviate threats to user privacy, ensure healthcare data confidentiality, address interoperability challenges, and efficiently manage privileges and access.

P47 [79] presents an architecture based on Solid and the Web of Things that enables controlled access to composite IoT devices and their data. It offers fine-grained access control, allowing users to interact with authorized composite service data stored in nested Linked Data Platform containers.

With this research question, we were particularly interested in understanding how PDS has been employed to handle privacy issues in the IoT domain. In this regard, we found only two studies (P35 and P49) that provided answers to this question.

First, in P35 [67], the authors introduced an IoT device recommendation approach that involves annotating devices using Linked Open Data resources. Semantic annotations are employed to calculate the similarity between annotated devices and make recommendations. As it is essential to access user data for this approach, privacy concerns are introduced; in order to mitigate these privacy issues, the authors adopted the Solid framework, enabling users to maintain control over the data used in the recommendation process.

Finally, P49 [81] presents an architecture including IoT devices as nodes in the Solid ecosystem, moving to a user-centric and decentralized IoT. The authors proposed Self-Sovereign Identity as an authentication mechanism for Solid.

### 4.3. PDS Solutions to Address Privacy Issues

To answer research question RQ3 (see Section 3), we examined five studies: P1, P5, P32, P34, and P48. Figure 7 depicts the findings from the perspective of using PDS to address privacy issues. In all, 60% of the findings concentrated on utilizing PDS to enhance user control, place users at the center of decision-making, and provide a foundational solution, while the remaining 40% focused on employing PDS to ensure compliance with regulations. Thus, based on these findings, we can answer this question by stating that PDS-based solutions for privacy issues focus on two main aspects, namely, user control and compliance with privacy regulations.

Below, we describe our findings (summarized in Table 10) using the following characteristics: (i) the perspective employed by the authors to apply PDS as a solution for privacy issues; (ii) the technology utilized in the proposal’s implementation; and (iii) the data domain.

P1 [34] proposed a new user-centric model for Personal Data (PD) management, addressing the dilemma of balancing the exploitation of PD and the development of more personalized applications while protecting individual privacy. The concept revolves around empowering individuals with control over the entire PD lifecycle, encompassing data acquisition, storage, processing, and sharing. In addition, the authors present PDS as a way to implement a user-centric model in which individuals can collect and store their own data. P5 [38] extended the PDS concept by introducing the My Data Store framework, a network of trusted applications that provides user control. It offers a set of tools to manage, control, and exploit heterogeneous PD collected by apps and sensors.

In study P32 [65], the authors adopted a wide-ranging perspective on PDS and critically examined its potential to empower individuals and tackle challenges within data processing ecosystems. Their primary focus was on data protection, encompassing the relationship between PDS and individuals’ rights, the legal foundations of data processing, and the impact of PDS on addressing information imbalances and online surveillance practices. Their findings revealed that despite the anticipated benefits of PDS, online data ecosystems continue face many challenges.

P34 [6] concentrated on digitizing privacy and data protection details, exploring the intricacies of implementing a service utilizing decentralized web technologies and semantic web standards and specifications. The primary aim was to streamline communication between data subjects and data controllers while ensuring compliance with GDPR regulations. The proposed service targeted pivotal challenges pertaining to GDPR-driven rights and obligations, facilitating negotiation of privacy terms, and managing access to personal data repositories.

P48 [80] introduces TIDAL, an implementation of a Solid application designed to promote seamless interactions between citizens and researchers in the context of health research. The system stores personal data in Solid pods via RDF, regulates access in order to query specific subsets of personal data, facilitates posting of both human-readable and machine-readable digital consent by researchers, and employs federated learning to analyze personal health data from numerous individuals.

## 5. Threats to Validity

In secondary research, the conclusions drawn and the chosen studies by researchers can significantly influence the outcomes [82]. Consequently, it becomes imperative to examine factors that can impact our study’s validity. In this section, we explore the potential threats to the validity of our research in order to ensure a credible interpretation of the results.

In this context, we envision three threats to the validity of the results of our systematic study: (i) bias in the selection of the included studies; (ii) the precision of the data extraction process; and (iii) the effective classification and interpretation of the data.

The selection process began by applying search strings to the research databases and identifying a preliminary set of studies based on their titles and abstracts. These papers were then compared against predefined inclusion and exclusion criteria (Section 3.1.3). The selected papers were fully read again and assessed against the established criteria. Subsequently, those articles meeting the inclusion criteria were chosen for further data extraction.

During data extraction, we efficiently addressed imprecision by leveraging search engine resources. We removed studies that lacked the keywords described in their title or abstract. Extraction was conducted collaboratively in pairs, with any discrepancies resolved through debate in order to ensure consensus.

Finally, we based our classification and interpretation on privacy threats in the IoT domain, paying special attention to PDS-based solutions.

## 6. Privacy: Open Challenges

Privacy is a recurring problem that is gaining increasing importance, mainly due the integration of smart devices into people’s daily lives. Furthermore, as observed in our review findings, the past five years have witnessed an increased number of publications, indicating growing interest in researching privacy solutions in the IoT context. Despite ongoing efforts, further research is required to address or mitigate the privacy concerns around IoT. In Table 11, we present the most common privacy issues found in the literature; even today, these remain open challenges, and new research is necessary to make progress on them. Below, we provide a brief description of these issues.

**Identification** 
The IoT system is inherently widespread, enabling devices to capture a wide array of data regarding users and their interactions with the environment. Service providers typically process these data beyond users’ control. Thus, a significant issue is the risk of identification, which involves linking personal data about an individual with an identifier such as a name or address. In the IoT domain, new technologies and the interconnection of various techniques elevate individuals’ susceptibility to identification threats.**Localization and tracking** 
are associated with specifying and recording an individual’s location through time and space using methods such as cell phone tracking, internet traffic analysis, or GPS data. The availability of extensive and comprehensive spatial and spatiotemporal data has sparked growing interest in leveraging geographic information and spatial analysis. As the IoT system advances, several factors amplify the challenges related to localization. These factors include the proliferation of location-aware applications, the enhanced precision and omnipresence of data collection technologies, and increased interaction with IoT devices which record the user’s location and activities.**Profiling** 
involves gathering and analyzing data about an individual’s activities and behaviors over extended periods in order to categorize them based on certain characteristics. This information is typically acquired without user consent and combined with other personal data to construct a more comprehensive profile. This approach is frequently employed for personalization in e-commerce, such as recommender systems, advertisements, and internal optimization tailored to customer demographics and interests.**Privacy-violating interaction and presentation** 
refers to the delivery of personal details through a public environment and their disclosure to an unwanted audience. Numerous IoT applications spanning manufacturing, infrastructure, healthcare systems, and more require extensive user connectivity. In such systems, it is plausible that information may be made available to users through smart devices in their environment. However, many interaction and presentation mechanisms are inherently public, which consequently creates privacy concerns when sensitive information is exchanged between the user and the system.**Lifecycle transitions** 
relates to the disclosure of private information when ownership of a consumer product changes throughout its lifecycle. Because consumer products that store private data, such as smartphones, cameras, and laptops, typically remain under the control of the same owner throughout their entire lifecycle, this issue is not frequently observed; however, with the increasing connectivity and inclusion of private data in a growing number of everyday objects, the risk of privacy breaches due to ownership changes is rising.**Inventory attacks** 
entail the unauthorized gathering of information regarding the presence and attributes of personal belongings. Typically, these attacks leverage the fingerprints of IoT devices, including parameters such as communication speed and response time. Supposing that the IoT’s potential is fully realized, all smart devices will be accessible via the internet, creating opportunities for unauthorized entities to exploit this and compile an inventory of items associated with a specific target. An inventory attack could be utilized for profiling individuals, as possession of specific items could reveal private information about the owner.**Linkage** 
relates to the uncontrolled disclosure of information due to the integration of previously isolated data sources. Combining various parts of an individual’s data may uncover new insights that the owner did not anticipate, and this revealed information is considered a breach of privacy. In the IoT context, the threat of linkage is amplified by the integration of diverse organizations, creating a more heterogeneous and widely distributed system.**Information leakage** 
IoT applications are interconnected via wireless communication, and potential attackers can intercept wireless signals to acquire transmitted information. With the widespread adoption of smartphones, people often need to share their identity and location in order to utilize their devices and to access relevant applications and essential services. If such sensitive data were disclosed or obtained by malicious actors in this way, it could pose significant privacy risks.

At the same time, in examining the outcomes of our review it becomes evident that there has been limited exploration of the connection between IoT and PDS as a method for addressing privacy risks. PDS enhances user control, allowing users to store, manage, and share personal data and digital assets while controlling access and utilization. For example, as proposed by Pinto and Prazeres [83], decentralized data processing can be utilized to place control over decisions regarding data treatment in the hands of each user. Their proposal embraces a Personal Data Store within an IoT/FoT architecture, positioning the user at the core of decision-making and empowering them to determine under what circumstances, by whom, and to what extent their data will be processed. Consequently, identification risks can be mitigated in two ways: (i) through anonymous interaction with services, and (ii) by the difficulty of re-identification, as users can decide the extent to which their data will be disclosed. Furthermore, risks related to profiling, linkage, localization, and tracking can be mitigated by fostering the benefit of user control provided by PDS. Thus, further research and development in this area is necessary.

## 7. Conclusions and Future Works

In this article, we have presented the outcomes of a systematic mapping review of the literature aimed at evaluating the current state-of-the-art research on privacy issues in the IoT domain, with a specific focus on the utilization of PDS as a privacy protection solution. We conducted the review following established best practices in the literature [14,32,33], systematically selecting 49 studies from an initial pool of 176 publications gathered from five search engines (ACM, IEEE Xplore, Science Direct, Scopus, and Wiley).

Based on our findings, we discovered many studies that have addressed privacy risks within the IoT context. The most common IoT privacy issues discussed in the literature include identification, localization and tracking, profiling, privacy-violating interaction and presentation, lifecycle transitions, inventory attacks, linkage, and information leakage.

Despite the worry about privacy risks, we have identified limited solutions to the problem, most of which have been introduced in the last five years, and especially in employing PDS as an alternative, indicating increasing concern regarding these issues. We identified only two works addressing privacy issues by employing PDS in this manner. This situation demonstrates that deeper investigation is necessary in order to understand the potential utility of PDS in this context.

As part of our future work, we aim to further develop our PDS-based solution [83] to address privacy issues in the context of Fog of Things [84] data [85]. We advocate for a user-centric approach, and assert that it can mitigate various privacy issues through an approach that enhances users’ control while improving transparency, awareness, and trust.

## Figures and Tables

**Figure 1 sensors-24-02197-f001:**
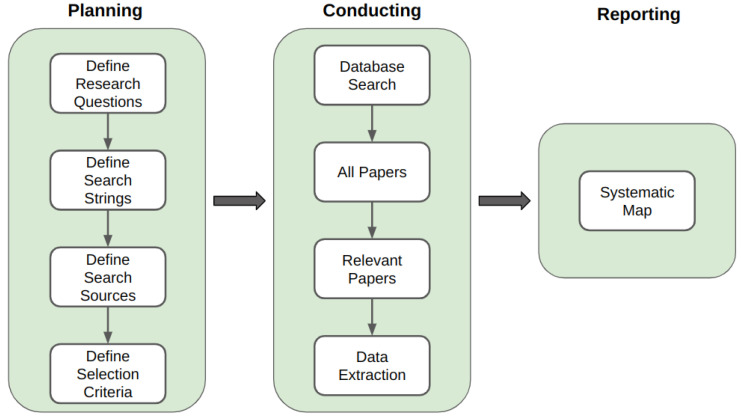
The systematic mapping process.

**Figure 2 sensors-24-02197-f002:**
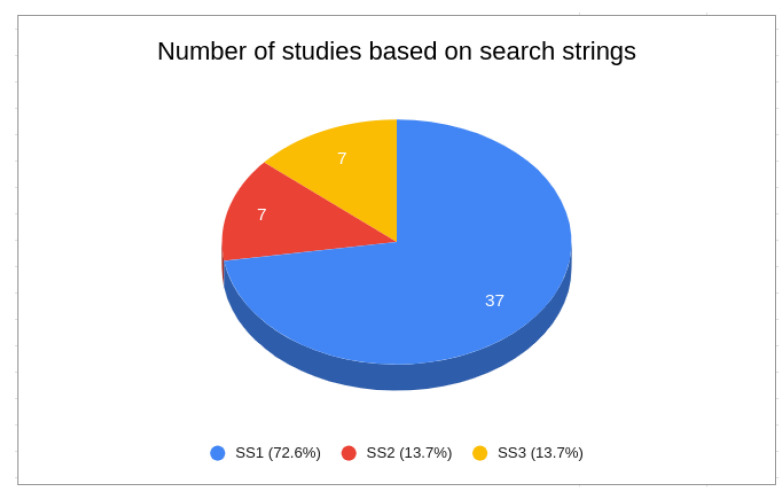
Number of studies based on search string.

**Figure 3 sensors-24-02197-f003:**
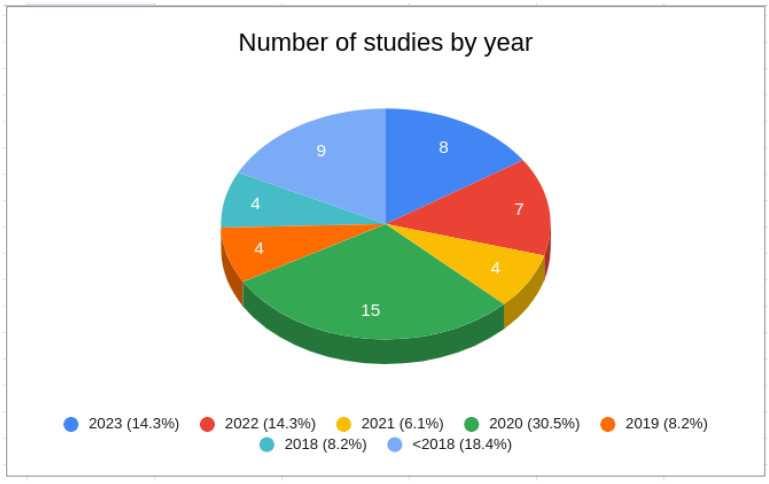
Number of studies by year.

**Figure 4 sensors-24-02197-f004:**
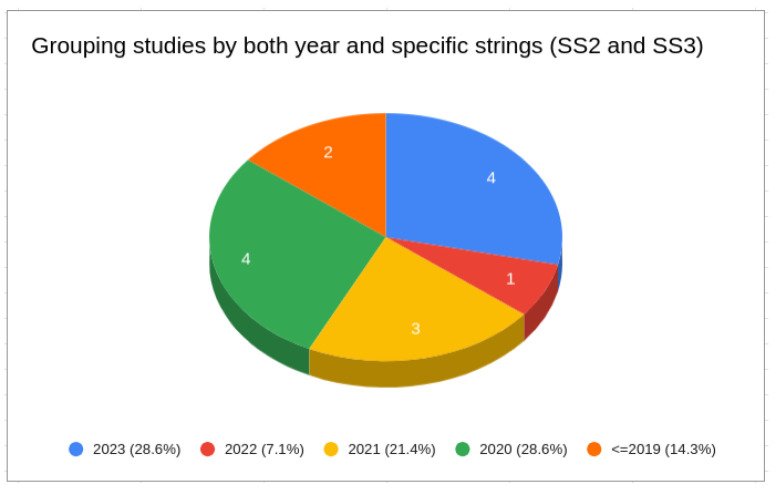
Studies grouped by both year and specific string (SS2 and SS3).

**Figure 5 sensors-24-02197-f005:**
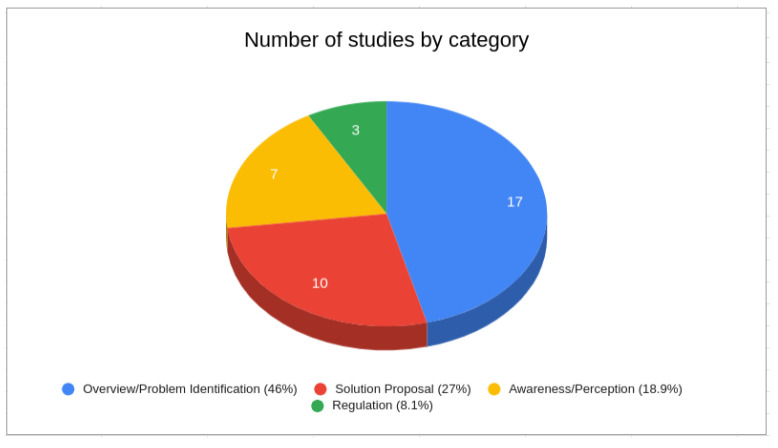
Categorization of SS1 results.

**Figure 6 sensors-24-02197-f006:**
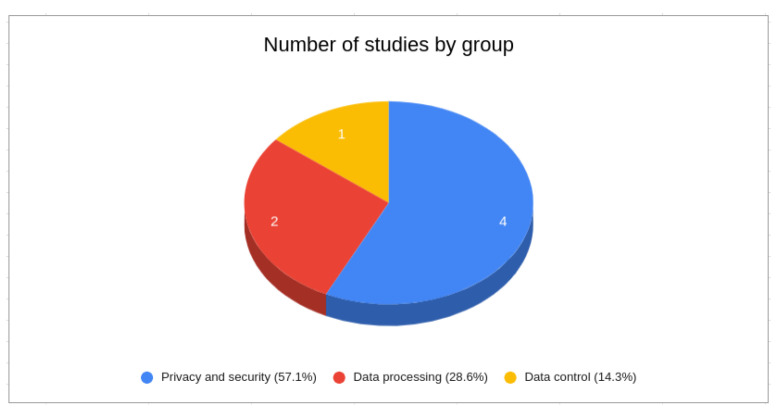
Categorization of SS2 results.

**Figure 7 sensors-24-02197-f007:**
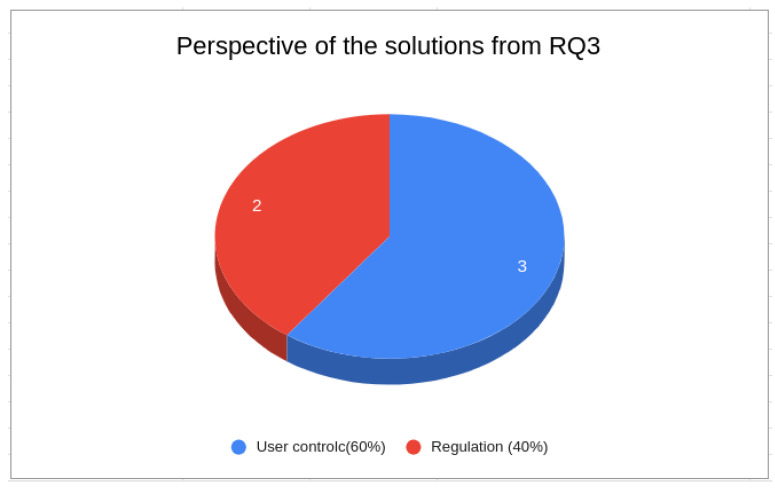
Perspectives of the solutions from RQ3.

**Table 1 sensors-24-02197-t001:** Comparison of related works.

Authors	Year	Systematic	Domain	Main Goal
Ogonji et al. [15]	2020	✓	IoT	Privacy and security threats, attack surface, vulnerabilities, and countermeasures in IoT.
Sarwar et al. [25]	2021	x	IoT and Fog	Privacy preservation in IoT and fog-enabled IoT
Zainuddin et al. [16]	2021	x	IoT Applications	Privacy threats in IoT applications
Islam et al. [17]	2022	✓	IoT and Big Data	Types of privacy attacks and solutions in the IoT and Big Data contexts
Zarandi et al. [20]	2022	x	Smart Farming	Big Data privacy in smart farming
Hireche et al. [21]	2022	x	IoMT	Privacy and security in IoMT
Shahid et al. [23]	2022	x	IoHT	Data breaches and legal regulations in IoHT
Alabdulatif et al. [24]	2023	x	IoNT	Overview of IoNT and its security and privacy issues
Torre et al. [18]	2023	✓	IoT Devices	Privacy-preservation techniques for IoT devices
Tokas et al. [26]	2023	x	IoT	Privacy-enhancing technologies and GDPR-specific privacy principles
Demertzi et al. [27]	2023	x	IIoT	Privacy requirements in the Industrial IoT
Fallatah et al. [28]	2023	x	PDS	PDS and its advantages and disadvantages
Zubaydi et al. [29]	2023	✓	IoT and Blockchain	Blockchain solutions for IoT security and privacy issues
Asqah and Moulahi [30]	2023	x	IoT and Blockchain	Security and privacy challenges arising from the integration of federated learning and blockchain within IoT
Kamalov et al. [22]	2023	✓	IoMT	Security concerns in the Internet of Medical Things
Rodriguez et al. [31]	2023	x	IoT	Solutions for IoT privacy based on machine learning and deep learning
Khan et al. [19]	2023	x	IoT	Privacy and security issues in IoT
This study	2024	✓	IoT and PDS	Privacy issues in IoT and solutions through personal data stores

**Table 2 sensors-24-02197-t002:** Research questions.

ID	Research Question (RQ)
RQ1	What are the main foci of the published studies regarding privacy within the IoT domain?
RQ2	What issues are addressed by Personal Data Stores in the IoT context?
RQ3	How have Personal Data Stores been used to address privacy issues in a general context?

**Table 3 sensors-24-02197-t003:** Search strings.

ID	Search String (SS)
SS1	(“Internet of things privacy” or “IoT privacy” or “fog computing privacy” or “edge computing privacy”) and (threat or risk)
SS2	(“Personal Data Store” or “Personal data vault” or “social linked data”) and (“Information privacy” or “data protection” or “data privacy”)
SS3	(“Personal Data Store” or “Personal data vault” or “social linked data”) and (“Internet of Things” or IoT or “Fog Computing” or “Edge Computing”)

**Table 4 sensors-24-02197-t004:** Research databases.

Name	URL
ACM Digital Library	https://dl.acm.org/ accessed on 16 August 2023
IEEE Xplore	https://ieeexplore.ieee.org/ accessed on 16 August 2023
Science Direct	https://sciencedirect.com/ accessed on 17 August 2023
Scopus	https://www.scopus.com/ accessed on 17 August 2023
Wiley	https://onlinelibrary.wiley.com/ accessed on 18 August 2023

**Table 5 sensors-24-02197-t005:** Number of studies per database/string.

Databases	SS1	SS2	SS3
ACM	3	0	1
IEEE Xplore	12	1	0
ScienceDirect	0	0	2
Scopus	71	9	14
Wiley	50	4	9

**Table 6 sensors-24-02197-t006:** Number of papers eliminated by selection criteria.

Selection Criteria	SS1	SS2	SS3
**SC1**	136	14	26
**SC2**	95	12	22
**SC3**	95	12	22
**SC4**	85	12	22
**SC5**	37	7	7

**Table 7 sensors-24-02197-t007:** Selected IoT privacy studies.

ID	Work	Title	Year	Item Type
**P1**	[34]	How do I manage my personal data? - A telco perspective	2012	Conference
**P2**	[35]	IoT-Privacy: To be private or not to be private	2014	Conference
**P3**	[36]	Privacy in the Internet of Things: threats and challenges	2014	Journal
**P4**	[37]	Internet of things: Privacy issues revisited	2015	Journal
**P5**	[38]	Building an eco-system of trusted services via user control and transparency on personal data	2015	Journal
**P6**	[39]	The Quest for Privacy in the Internet of Things	2016	Journal
**P7**	[40]	Privacy and security: Key requirements for sustainable IoT growth	2017	Conference
**P8**	[41]	Research on IoT Privacy Security Risks	2017	Conference
**P9**	[42]	Privacy preserving Internet of Things: From privacy techniques to a blueprint architecture and efficient implementation	2017	Journal
**P10**	[43]	User perceptions of smart home IoT privacy	2018	Journal
**P11**	[44]	Towards a standard-based security and privacy of IoT system’s services	2018	Conference
**P12**	[45]	IoT Privacy in 5G Networks	2018	Conference
**P13**	[46]	Poster abstract: Preserving IoT privacy in sharing economy via smart contract	2018	Conference
**P14**	[47]	Internet of Things (IoT): Security and Privacy Threats	2019	Conference
**P15**	[48]	A survey on internet of things and cloud computing for healthcare	2019	Journal
**P16**	[49]	A brief survey on IoT privacy: Taxonomy, issues and future trends	2019	Journal
**P17**	[50]	The internet of things privacy	2019	Journal
**P18**	[51]	Bringing Privacy Control Back to Citizens: DISPEL — a Distributed Privacy Management Platform for the Internet of Things	2020	Conference
**P19**	[52]	Ask the Experts: What Should Be on an IoT Privacy and Security Label?	2020	Conference
**P20**	[53]	An Overview of Privacy Issues in IoT Environments	2020	Conference
**P21**	[54]	A survey-based exploration of users’ awareness and their willingness to protect their data with smart objects	2020	Journal
**P22**	[55]	The internet of things: Multi-faceted research perspectives	2020	Journal
**P23**	[56]	Semantic-based privacy settings negotiation and management	2020	Journal
**P24**	[57]	Presence metadata in the Internet of Things: Challenges and opportunities	2020	Conference
**P25**	[58]	Towards a heterogeneous IoT privacy architecture	2020	Conference
**P26**	[59]	IoT privacy and security: Challenges and solutions	2020	Journal
**P27**	[60]	A Framework for Privacy Policy Compliance in the Internet of Things	2020	Journal
**P28**	[61]	IoT Security, Privacy, Safety and Ethics	2020	Journal
**P29**	[62]	PDSproxy++: Proactive proxy deployment for confidential ad-hoc personalization of AI services	2020	Conference
**P30**	[63]	PDSProxy: Trusted IoT Proxies for Confidential Ad-hoc Personalization of AI Services	2020	Conference
**P31**	[64]	SOLIOT-Decentralized data control and interactions for IoT	2020	Journal
**P32**	[65]	Personal information management systems: A user-centric privacy Utopia?	2020	Journal
**P33**	[66]	Systematically Quantifying IoT Privacy Leakage in Mobile Networks	2021	Journal
**P34**	[6]	Challenges in the Digital Representation of Privacy Terms	2021	Journal
**P35**	[67]	Towards a Privacy Conserved and Linked Open Data Based Device Recommendation in IoT	2021	Journal
**P36**	[68]	Large-Scale Multiobjective Federated Neuroevolution for Privacy and Security in the Internet of Things	2022	Journal
**P37**	[69]	TrafficSpy: Disaggregating VPN-encrypted IoT Network Traffic for User Privacy Inference	2022	Conference
**P38**	[70]	On the Data Privacy, Security, and Risk Postures of IoT Mobile Companion Apps	2022	Journal
**P39**	[71]	Promoting Information Privacy Protection Awareness for Internet of Things (IoT)	2022	Journal
**P40**	[72]	A Two-Fold Study to Investigate Users’ Perception of IoT Information Sensitivity Levels and Their Willingness to Share the Information	2022	Journal
**P41**	[73]	Consumer IoT and its under-regulation: Findings from an Australian study	2022	Journal
**P42**	[74]	SHARIF: Solid Pod-Based Secured Healthcare Information Storage and Exchange Solution in Internet of Things	2022	Journal
**P43**	[75]	Protected or Porous: A Comparative Analysis of Threat Detection Capability of IoT Safeguards	2023	Conference
**P44**	[76]	The Awareness of Internet of Things (IoT) Privacy Risk: A Survey Study	2023	Journal
**P45**	[77]	Privacy and Ethical Considerations of Smart Environments: A Philosophical Approach on Smart Meters	2023	Journal
**P46**	[78]	Knowledge elicitation methodology for evaluation of Internet of Things privacy characteristics in smart cities	2023	Journal
**P47**	[79]	A Solid Architecture for Machine Data Exchange with Access Control	2023	Conference
**P48**	[80]	Analyze Decentralized Personal Health Data using Solid, Digital Consent, and Federated Learning	2023	Conference
**P49**	[81]	A Decentralized Smart City Using Solid and Self-Sovereign Identity	2023	Journal

**Table 8 sensors-24-02197-t008:** Summary of privacy risks and related studies.

Reference	P2 [36]	P4 [39]	P5 [40]	P6 [41]	P12 [47]	P14 [49]	P17 [53]	P21 [57]	P23 [59]	P25 [61]	P28 [69]
Identification	✓		✓		✓	✓	✓	✓		✓	
Localization and tracking	✓	✓			✓	✓			✓	✓	
Profiling	✓	✓			✓					✓	
Interaction and presentation	✓				✓						
Lifecycle transitions	✓				✓					✓	
Inventory attack	✓				✓					✓	
Linkage	✓				✓					✓	
Information leakage				✓			✓				✓

**Table 9 sensors-24-02197-t009:** Comparison of findings from RQ2.

Reference	Challenge	PDS	IoT Scenario
P29 [62]	Initialization overhead	PDSProxy	AI services for IoT devices
P30 [63]	Secure data transmission	PDSProxy	AI services for IoT devices
P31 [64]	Constraint devices, battery-powered sensors, and network disruptions	Solid	Industrial applications
P42 [74]	Data communication	Solid	Healthcare
P47 [79]	Access control to IoT devices and data	Solid	IoT devices
P35 [67]	Privacy	Solid	IoT device recommendation
P49 [81]	Privacy	Solid	IoT devices

**Table 10 sensors-24-02197-t010:** Comparison of the findings from RQ3.

Reference	Perspective	Technology	Domain
P1 [34]	User control	x	General personal data
P5 [38]	User control	My Data Store	Personal data from apps and sensors
P32 [65]	Regulation	x	General personal data
P34 [6]	Regulation	x	General personal data
P48 [80]	User control	Solid	Health personal data

**Table 11 sensors-24-02197-t011:** Privacy challenges.

Challenge	Description
Identification	To link personal data about an individual with an identifier, such as a name or address
Localization and tracking	To specify and record an individual’s location through time and space
Profiling	To gather and analyze data about an individual’s activities and behaviors over extended periods in order to categorize them.
Interaction and presentation	Refers to the act of sharing personal details in a public environment and subsequently disclosing them to an unintended audience.
Lifecycle transitions	Refers to the disclosure of private information when ownership of a consumer product changes throughout its lifecycle
Inventory attacks	To gather information regarding the presence and attributes of personal belongings
Linkage	Disclosure of information resulting from the integration of previously isolated data sources.
Information leakage	The disclosure or obtaining of data by malicious actors

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
