# Peer review of "A Systematic Review on Privacy-Aware IoT Personal Data Stores"

_sensors, 2024, doi:10.3390/s24072197_

Round 1

Reviewer 1 Report

Comments and Suggestions for Authors

The manuscript is well-written and the methodology sounds, however, I have the following concerns:

Novelty: with a quick search on Google Scholar, many similar studies could be identified. Just a few of them are listed below. So, the manuscript does not have high novelty.

https://www.techrxiv.org/doi/full/10.36227/techrxiv.19874329.v1

https://dl.acm.org/doi/abs/10.1145/3648480

https://link.springer.com/chapter/10.1007/978-3-319-69462-7_35

Related works

Many related studies have not been included! Why?

Results

The results and future directions are missing the minimum visualization. You may consider adding one or two diagrams to enhance the readability of the paper. Also, you may add one or two tables to summarize the major findings in Section 4 and Section 5.

Comments on the Quality of English Language

Minor corrections required

Author Response

Please, see attached letter.

Reviewer 2 Report

Comments and Suggestions for Authors

The author introduced the A Systematic Review on Privacy-aware IoT Personal Data Stores. However, I have some suggestions like

1. The author missed discussing cryptography and access control based related works in the introduction and literature survey sections. Please refer the following paper for  https://doi.org/10.3390/app13063970

2. The author has to mentions the challenges and limitations in the form of tabular format. 

3. Mentioned the features of proposed methods

4. The author missed the proposed method limitations. 

Comments on the Quality of English Language

The moderate language editing is required

Author Response

Please, see attached letter.

Reviewer 3 Report

Comments and Suggestions for Authors

Abstract: 

- The abstract states that the review is focused on 'Privacy issues and solutions in the context of the IoT', whereas the title focuses only on Personal Data Stores as a solution. Clarify whether the focus is on reviewing many solutions or just PDS.

- Highlight any important findings/conclusions that were identified with this review (e.g. any unexpected privacy issues?)

Introduction:

- line 16: 'leveraged generation' -- please explain 'leveraged' in this context. Do you mean that companies leverage data?

- lines 34-35: 'Such events contradict Westin’s privacy definition.' -- They do not contradict the definition. Perhaps you mean that they violate privacy, as defined by Westin.

- lines 54-55: 'It means an inaccurate understating of the data privacy concept.' -- This is not clear. Do you mean 'understanding' instead of 'understating'?

- lines 75-85: The bullet list of contributions should be more focused on the results, not on the process for finding the results. For example, 'systematically present 49 studies' is not a research contribution by itself, but the open challenges you found, or any other conclusions, are a major contribution (so please list the main findings/conclusions in one of the bullet points). As another example, if you were able to group/categorize these studies in a way that is useful for other researchers, please mention that as a bullet point.

Related Works:

- lines 144-145: 'In conclusion, our search revealed a dearth of prior research addressing our primary research question, confirming the necessity for ...' -- It is not clear which one is your primary research question. I assume it is RQ1 - 'What are the main foci of the published studies regarding privacy within the IoT domain?'

How does it follow from Section 2 that there is a dearth of research addressing RQ1? You list 6 literature reviews. What is missing from them that justifies conducting another review?

Research Method:

- Table 2 and lines 166-175: I do not understand the difference between RQ2, RQ3, and RQ4. They all seem to be variants of asking 'How are PDS used to solve IoT privacy issues?'. Except possibly RQ3, which seems to be about a more general context, not just IoT, but then I don't see how non-IoT contexts are relevant for this survey. Please explain the differences, or merge some of the RQs together.

- The groups identified in Figure 5 are a useful way to categorise privacy papers. Occasionally there are papers that talk about privacy in an economic/business context (e.g. monetizing personal data, how much is privacy worth), but perhaps there aren't many of this type focusing explicitly on IoT data.

Results:

- lines 265-267: The contrast between security and privacy is interesting to note, but it should be explained more. For example, PDS aim to protect privacy via access control, but access control can also be seen as a security solution. Please provide some examples where a security solution was proposed but it did not address privacy.

- Table 8: This table is useful. A famous paper on privacy risks (from a legal perspective, and not specific to IoT) is Solove, D. J. (2006). A Taxonomy of Privacy. University of Pennsylvania Law Review, 154(3), 477–564. https://doi.org/10.2307/40041279

Privacy Open Challenges:

- The short descriptions of each issue are useful, but it seems they could have been written even without conducting the whole review. It is not clear how these are open challenges, i.e. why they have not been adequately addressed by the existing research and what exactly remains to be done. As the focus of this review is also on PDS, it would be useful to expand the last paragraph of Section 5 (lines 539-543) to discuss how PDS can address each of the issues, i.e. what is the unique promise that PDS can make about solving these issues, that other proposed solutions cannot make?

Comments on the Quality of English Language

Please see comments above.

Author Response

Please, see attached letter.

Reviewer 4 Report

Comments and Suggestions for Authors

The work "A Systematic Review on Privacy-aware IoT Personal Data Stores" contains all necessary parts of such type of submissions. Authors defined research questions, search strings, databases, and selection criteria.

It is well-written, contains comparison with other systematic reviews, and description of all selected and investigated manuscripts. In addition to the analysis of research questions, the authors also introduced open challenges, provided conclusions and future work directions.

In my opinion, the following additions will help to improve the work done:

1. In Section 4 it is required to not only describe selected studies, but also to provide concise answers to each research question, extracted from the analysis of studies.

2. In Section 5, it is important to give more insights about the role of privacy-aware IoT personal data stores in the mentioned open challenges. Is it a solution? Or we also need other countermeasures?

Author Response

Please, see attached letter.

Round 2

Reviewer 1 Report

Comments and Suggestions for Authors

The authors have addressed several of the previous feedback points. However, the main concern is still the Novelty and the literature. I suggest you discuss your paper's significance in the introduction adding more references to support it. Also, I suggest discussing at least another 10 new references from 2023 and 2024. You can find many in MDPI database.

Comments on the Quality of English Language

Minor
